# Study protocol of the PEruvian Registry of ST-segment Elevation Myocardial Infarction II (PERSTEMI-II) study

Manuel Chacón-Díaz[1], Akram Hernández-Vásquez[2], Rodrigo Vargas-Fernández[3]*, Guido Bendezu-Quispe[4]

1 Instituto Nacional Cardiovascular, EsSalud, Lima, Peru, 2 Vicerrectorado de Investigación, Centro de Excelencia en Investigaciones Económicas y Sociales en Salud, Universidad San Ignacio de Loyola, Lima, Peru, 3 Universidad Científica del Sur, Lima, Peru, 4 Centro de Investigación Epidemiológica en Salud Global, Universidad Privada Norbert Wiener, Lima, Peru

* jvargasf@cientifica.edu.pe

## Abstract

### Background

Myocardial infarction (MI) is the most prevalent cardiovascular disease globally and is considered a public health problem. In Peru, MI is the second leading cause of death at the national level, with a mortality rate that exceeds 10% in the hospital setting. The study aims to determine the clinical and epidemiological characteristics of ST-segment elevation myocardial infarction (STEMI) in tertiary care facilities belonging to the Peruvian public health system.

### Methods and analysis

This will be a prospective, observational, multicenter study, with baseline and two follow-up assessments: at admission to the health service, and 30 days and 12 months after admission. This multicenter study will be conducted in 27 hospitals located in the main cities of Peru. The patients included in the study will be over 18 years of age, of either sex, and will have been admitted to the health facility with a diagnosis of acute coronary syndrome with ST-segment elevation. The Kaplan-Meier method will be used to estimate the cumulative in-hospital mortality of patients at 30 days and 12 months of follow-up, and the log-rank test will be used to evaluate the differences between the survival curves between reperfused and non-reperfused patients. Subsequently, to evaluate the risk factors for successful reperfusion and cardiovascular adverse events, generalized linear models of the binomial family with log link function will be used to estimate the bivariate and multivariate relative risk (RR) with their respective 95% confidence intervals. This project was approved by the Ethics and Research Committee of the National Cardiovascular Institute (Instituto Nacional Cardiovascular "Carlos Alberto Peschiera Carrillo"—INCOR [in Spanish]; Approval report 21/2019-CEI).

**Funding:** The authors received no specific funding for this work.

**Competing interests:** The authors have declared that no competing interests exist.

## Discussion

Among the strengths, the observational design will allow the inclusion of a large sample of patients, which will significantly contribute to the knowledge base on STEMI in Peru. It should be noted that this study is the first to examine the clinical-epidemiological characteristics of STEMI in high-resolution hospital centers in Peru with follow-up one year after the event, providing knowledge of these observable characteristics in daily clinical routine. Likewise, the multicenter nature of the study will increase the external validity of the findings. In terms of limitations, the observational design of the study can only describe associations and not causality. Furthermore, since data from medical records will be used, there could be imprecision in the data.

## Introduction

Cardiovascular diseases (CVD) are the leading cause of death in the world, with approximately 17.8 million deaths in 2017 being due to this cause, and more than 75% of these deaths occur in middle and low-income countries [1]. Within CVD, myocardial infarction (MI) is the most frequent disease, affecting 1.72% of the world population, with a prevalence of 1655 people with MI per 100 000 and a projected increase in the coming years [2], constituting a major public health problem.

Regarding the presentation of MI, ST-segment elevation myocardial infarction (STEMI) is one of the clinical forms of presentation of MI (the others are non-ST-segment elevation MI and unstable angina) [3]. STEMI is characterized by symptoms characteristic of myocardial ischemia with ST-segment elevation in the electrocardiogram and with at least one value above the 99th percentile of the upper reference limit [4]. This form of presentation of MI is associated with high morbidity and mortality. Among hospital admitted patients in Europe, the approximate incidence of STEMI is 43-144 cases per 100,000 inhabitants per year [5], with a pattern of being relatively more frequent in the young than in the elderly and in males than in females [6] and with in-hospital mortality of between 3 and 13.5% [7, 8]. In the United States, the reported adjusted incidence of STEMI is about 50 per 100,000 [9], with in-hospital mortality of 5-18% per year in unselected patients [10].

Regarding the use of reperfusion in STEMI, data from the CRUSADE registry described that the percentage of reperfusion in the United States is 82.5%, and up to 7.2% do not receive reperfusion despite being indicated [10], while in Europe, it has recently been reported that reperfusion in STEMI is predominantly by percutaneous coronary surgery (72.2%) followed by fibrinolysis (18%), and 9% do not receive reperfusion [11]. It should be noted that the quality of hospital care received by patients with STEMI or non-ST-segment elevation myocardial infarction (NSTEMI) can reduce a large proportion of disparities in access to treatment, time to intervention, and cardiac rehabilitation. In this regard, updates have been made of assessments of hospital quality of care measures, highlighting seven measures (four of clinical effectiveness and three of patient safety) that address STEMI and NSTEMI patients [12]. Particularly, in comatose STEMI patients with out-of-hospital cardiac arrest (as one of the specific clinical effectiveness measures) and in patients with cardiac arrest due to ventricular tachycardia and ventricular fibrillation, therapeutic hypothermia is considered and should be initiated as soon as possible [12]. Regarding quality measures related to patient safety, it is observed that the inappropriate use of nonsteroidal anti-inflammatory drugs, prasugrel at

discharge in patients with a previous history of stroke or transient ischemic attack (TIA), and high doses of aspirin with ticagrelor at discharge are measures that should be advised in the management of patients with both types of myocardial infarction and may reduce the risks of treatment failure [12].

In the Latin American and the Caribbean region, there are few reports on the prevalence of STEMI, with this presentation accounting for 40% of acute coronary syndromes in Argentina [11] and 34.8% in Mexico [12]. In Peru, chronic diseases are the leading cause of death and burden of disease [13, 14]. In recent decades, MI is the second cause of death in Peru (with an increase of 29.9% between 2009 and 2019), and is only surpassed by lower respiratory tract diseases [13]. Concerning STEMI in Peru, the objective of a previous study conducted in public and private hospitals of high-level specialization was to identify the epidemiological characteristics of STEMI [15]. This study found that this disease presents in-hospital mortality of 10.1%, and the main cause of death was cardiogenic shock, which most frequently affects males between 60 and 70 years of age. In relation to therapeutic management, fibrinolysis was used in 38% of cases, primary angioplasty was performed in 29%, 33% did not receive reperfusion during the first 12 hours of infarct evolution, and a pharmacoinvasive strategy was used in 12.9% of patients [15]. However, to date, there are no medium- and long-term follow-up studies available on the epidemiological characteristics of the Peruvian population with STEMI.

In recent years, there has been evidence of an increase in cardiovascular risk factors in Peru, as well as changes in the management and care of patients with STEMI [16, 17]. This points to the need to update data on the clinical and epidemiological profile of STEMI in Peru, as well as obtain knowledge about the therapeutic management offered for this disease in this country. Thus, the research problem of this second national registry is: What are the clinical and epidemiological characteristics of STEMI in Peru, and what is its therapeutic management? Information on these aspects of STEMI are necessary and will serve as input for the evaluation and development of strategies in order to achieve better outcomes for patients with this medical condition.

## Materials and methods

### Study objective

The aim of the study is to determine the clinical and epidemiological characteristics of STEMI in tertiary care facilities belonging to the Peruvian public health system (Ministry of Health [MINSA in Spanish] and Social Health Insurance [EsSalud in Spanish]). This will allow the determination of the epidemiological profile of STEMI in Peru, including the identification of the most commonly used reperfusion strategies and their mode of application. Likewise, the aim is to measure the incidence of STEMI complications and identify the main complications, and determine the incidence and risk factors of cardiovascular adverse events derived from STEMI at 30 days and 1 year of follow-up.

### Study design and settings

This is a prospective, observational, multicenter study with assessments at baseline on admission to the health service and with 2 follow-up visits: at 30 days and 12 months.

This multicenter study will be conducted in 27 hospitals located in the main cities of Peru: Hospital Nacional Arzobispo Loayza, Hospital Nacional Cayetano Heredia, Hospital Nacional Dos de Mayo, Hospital Nacional HIpolito Unanue, Hospital Santa Rosa, Hospital MINSA Huancayo, Hospital MINSA Cajamarca, Hospital MINSA Cusco, Hospital MINSA Chiclayo, Instituto Nacional Cardiovascular (INCOR), Hospital Nacional Edgardo Rebagliatti Martins, Hospital Nacional Guillermo Almenara Irigoyen, Hospital Nacional Alberto Sabogal

Sologuren, Hospital Essalud Cusco, Hospital Essalud Arequipa, Hospital Essalud Ica, Hospital Essalud Chiclayo, Hospital Essalud Trujillo, Hospital Essalud Juliaca, Hospital Essalud Iquitos, Hospital Essalud Tumbes, Hospital Essalud Huaraz, Hospital Essalud Piura, Hospital MINSA Arequipa, Hospital MINSA Iquitos, Hospital Essalud Tacna, and Hospital Regional Lambayeque.

## Participants and selection criteria

Patients with STEMI attending MINSA and EsSalud tertiary health care facilities during 2020 will be invited to participate in the study. For the facilities to be considered for the study, they must have at least one cardiologist attending patients who agree to participate in the study as well as an intensive care unit. The patient inclusion criteria will be age greater than 18 years, male or female, admission to the health facility with a diagnosis of acute coronary syndrome with ST-segment elevation according to the fourth universal definition of MI [4]. Patients with a coronary syndrome without ST-segment elevation at the time of admission to the health facility, with non-persistent STEMI, and patients with Takotsubo syndrome will be excluded.

Withdrawal criteria: Patients voluntarily requesting to withdraw from the study. Patients lost to follow-up.

## Informed consent

Patients will be asked to provide consent by means of a written, signed, and dated IRB-approved consent form.

## Study procedures

A cardiologist from each participating center will be in charge of data collection and sending the data to the registry coordination center (located in the city of Lima).

Prior to the collection of patient data, informed consent will be obtained. The data collection process will be carried out in digital format. For this purpose, a website exclusively designed for data collection will be created. The data of the patients included in the study will be obtained during hospitalization and will be sent electronically by the responsible physician to the registry coordination center. The data registry system will only be accessible with the use of an electronic user ID and password. All data will be stored in a secure electronic database. Each center will only have access to the data they report during their participation in the study.

For each patient included, we will record the general characteristics (age, sex, care center), epidemiological characteristics (history of cardiovascular risk factors), disease presentation (symptoms, electrocardiogram characteristics, Killip Kimbal classification), access to reperfusion, type of reperfusion, time from first medical contact and time from ischemia to reperfusion, in-hospital treatment and discharge, in-hospital complications and mortality. Follow-up of the patients included in the registry will be carried out by telephone and in person at 30 days and one year after inclusion in the study. Follow-up data at 30 days and at one year will be collected by telephone and by reviewing the medical records of the patients. Detailed information on the variables to be collected can be found in S1 Table.

## Sample size and power

All patients fulfilling the inclusion criteria and who attended the centers participating in the study over the course of one year will be included. While there is no limit to the number of participants enrolled, a representative sample size has been calculated according to the in-

hospital mortality reported in the PERSTEMI study. Thus, a minimum sample size of 164 participants has been estimated, based on an α-level of 0.05, a 95% confidence interval (95% CI), a 10% loss to follow-up, and in-hospital mortality equal to 10.8% [15].

## Outcome measures

In-hospital mortality (cardiovascular and noncardiovascular).

Mortality at 30 days (cardiovascular and noncardiovascular).

One-year cardiovascular mortality.

In-hospital symptomatic heart failure (heart failure symptoms and/or signs [dyspnea, pulmonary rales, peripheral edema, third and fourth heart sounds] associated or not with a reduced ejection fraction less than 40%) [5].

## Statistical and analytical plans

The data obtained will be compiled in a spreadsheet using Microsoft Excel 2013 software (Microsoft Corporation, USA) and will subsequently be exported to Stata v.14.2 software (StataCorp LP, College Station, Texas, USA). Descriptive analysis will be performed to characterize the study participants. To do this, patients will be compared to the event with the covariates of interest described previously. Numerical variables will be compared by means and medians, and categorical variables by absolute frequency distributions and percentages. First, the Kaplan-Meier method will be used to estimate the cumulative in-hospital mortality of the patients at 30 days and 12 months of follow-up, and the log-rank test will be used to evaluate the differences between the survival curves between the type of reperfusion. To perform the Kaplan-Meier method, three assumptions must be met: 1) censoring time is independent of survival time, 2) the survival probabilities of the subjects are the same during follow-up, and 3) the event occurs at a specific time. In addition, univariate and multivariate Cox regression analysis will be performed to estimate hazard ratios (HR) and their respective 95% confidence intervals (CI). Cox regression is a semi-parametric model that assumes proportional hazards and estimates and infers the HR and baseline hazard and survival functions.

On the other hand, to evaluate the risk factors for successful reperfusion and cardiovascular adverse events, bivariate analysis of categorical variables will be performed using the chi-square test, and for numerical variables, the Student's t-test or Mann-Whitney U test will be used according to compliance with the normal distribution assumption. Likewise, generalized linear models of the binomial family with log link function will be used to estimate the bivariate and multivariate relative risk (RR) and their respective 95% CI. The factors that will be taken into account include: general characteristics (age, sex, center of care), epidemiological (pathological history of cardiovascular risk factors), disease presentation (symptoms, electrocardiogram characteristics, Killip Kimbal classification), access to reperfusion, type of reperfusion, first medical contact times and time from ischemia to reperfusion, in-hospital and discharge treatment.

Two-tailed p values less than 0.05 will be deemed statistically significant.

## Data storage and management

All participant data from each research site will be entered by the site principal investigator (PI), a position that corresponds to the cardiologist in charge of the research at the tertiary

health facilities participating in the study. For this process, an electronic registry system specifically developed for this study will be used (http://40.77.71.10/www/Perstemi2/). This system has been designed to record data of the study participants such as medical history and risk factors, electrocardiography results, characteristics of the clinical presentation of the disease, type, and characteristics of the reperfusion therapy indicated, complementary therapy, characteristics of the coronary revascularization surgery indicated, hospital therapy, medical therapy at discharge and safety (in-hospital events). The system has data quality control measures to identify missing data, outliers, and discrepancies at the time of recording and periodically. The data recorded in the database can only be accessed by PIs at their respective research site. Once the data have been entered, a unique identification code will be generated to preserve the anonymity of the patients whose data have been recorded. The computers used to record and store the data will have a username and password and will have limited access, and the general public will not have access to these computers. The PI of the study will have access to the final study data set.

## Ethics and dissemination

The study has been approved by the Ethics Committee of the "Instituto Nacional Cardiovascular "Carlos Alberto Peschiera Carrillo—INCOR" (in Spanish), a recognized Institutional Review Board by the Peruvian National Institute of Health (https://www.ins.gob.pe/registroEC/listaregistroCIEI.asp), with the Approval Report N˚ 21/2019-CEI of December 9, 2019. The study will be performed in accordance with the Declaration of Helsinki. This approval was obtained prior to the collection of data from the first patient participating in the study.

The results of the study will be disseminated in articles in peer-reviewed scientific journals and in abstract format at scientific events. To improve transparency in the reporting of the results, this study will follow the guidelines of the Strengthening the Reporting of Observational Studies in Epidemiology Statement (STROBE) [18] for the writing of articles generated from the study.

## Status and timeline of the study

Recruitment of the participants began on January 1, 2020, and was completed on December 31, 2020, and the follow-up is expected to be completed by December 2021. Preliminary analysis of in-hospital mortality will be conducted in 2021, and the final manuscript with the follow-up at one year will be completed by May 2022.

## Discussion

The study will provide information on the factors associated with STEMI reperfusion in the Peruvian population and the main reasons for not performing this procedure. It will also help to estimate in-hospital mortality at 30 days and one year after admission to the health facility. In this way, we seek to contribute to the knowledge of the clinical and epidemiological characteristics of STEMI in the public hospitals of MINSA and EsSalud in Peru, which attend the greatest number of patients with this diagnosis, and management is governed by clinical practice guidelines based on current scientific evidence.

In Peru, few multicenter studies have been conducted on the epidemiological and clinical characteristics of MI. Of note are the study by the National Registry of Acute Myocardial Infarction (RENIMA in Spanish) in 2006 [19] and RENIMA II in 2010 [20]. These two studies were conducted in hospitals of the Peruvian health system and reported the demographic and clinical outcomes of acute MI as well as the therapeutic management of the patients. Additionally, the PEruvian Registry of ST-segment Elevation Myocardial Infarction (PERSTEMI)

was conducted in 2017 [15], providing an update of information on the Peruvian population with STEMI in the health system. However, in the context of the current pandemic, an update of the PERSTEMI information is needed, allowing better approximations of the Peruvian reality within the context of a health emergency.

Among the strengths of this study, the observational design of the study will allow the inclusion of a large sample of patients, which will significantly contribute to the knowledge base on STEMI in Peru. This study is the first to examine the clinical-epidemiological characteristics of STEMI in high-resolution hospital centers in Peru with a follow-up one year after the event, providing relevant information about these observable characteristics in daily clinical routine. Furthermore, the multicenter nature of the study will increase the external validity of the findings. However, regarding the limitations of the study, the observational design of the study can only describe associations and not causality. Moreover, there could be imprecision in the data analyzed since data from medical records will be used.

Currently, the SARS-COV-2 pandemic presents multiple research challenges, especially in relation to studies that require adequate and timely data collection. In particular, this health crisis poses threats to the execution of research studies due to problems in external validity, replicability of studies, quality of the data collected, and recruitment of participants [21, 22]. In relation to recruitment, there are several problems in interviewing and assessing participants, which in the current context have been partially solved by flexibility and virtual solutions. However, these solutions are difficult in conditions requiring physical assessment. Therefore, the present study has been conducted within the context of the pandemic and under the limited conditions of access to health facilities, without compromising the quality of the data or the recruitment of participants.

## Supporting information

**S1 Table. List of variables to be collected.**
(DOCX)

## Acknowledgments

To all PERSTEMI II registry investigators, residents, and support staff who, despite a difficult year, have actively collaborated in the study. The authors are grateful to Donna Pringle for reviewing the language and style.

## Author Contributions

**Conceptualization:** Manuel Chacón-Díaz.

**Investigation:** Manuel Chacón-Díaz, Akram Hernández-Vásquez, Rodrigo Vargas-Fernández, Guido Bendezu-Quispe.

**Methodology:** Akram Hernández-Vásquez, Rodrigo Vargas-Fernández, Guido Bendezu-Quispe.

**Project administration:** Manuel Chacón-Díaz.

**Resources:** Manuel Chacón-Díaz.

**Supervision:** Manuel Chacón-Díaz, Akram Hernández-Vásquez.

**Writing – original draft:** Manuel Chacón-Díaz, Akram Hernández-Vásquez, Rodrigo Vargas-Fernández, Guido Bendezu-Quispe.

**Writing – review & editing:** Manuel Chacón-Díaz, Akram Hernández-Vásquez, Rodrigo Vargas-Fernández, Guido Bendezu-Quispe.

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
