## [Decision Letter · Decision Letter 0]

25 Aug 2021

PONE-D-21-11641

Study protocol of the PEruvian Registry of ST-segment Elevation Myocardial Infarction II (PERSTEMI-II) study

PLOS ONE

Dear Dr. Vargas-Fernández,

Thank you for submitting your manuscript to PLOS ONE. After careful consideration, we feel that it has merit but does not fully meet PLOS ONE’s publication criteria as it currently stands. Therefore, we invite you to submit a revised version of the manuscript that addresses the points raised during the review process.

We look forward to receiving your revised manuscript.

Kind regards,

Chiara Lazzeri

Academic Editor

PLOS ONE

Journal Requirements:

Reviewers' comments:

Reviewer's Responses to Questions

**Comments to the Author**

1. Does the manuscript provide a valid rationale for the proposed study, with clearly identified and justified research questions?

Reviewer #1: Yes

Reviewer #2: Yes

2. Is the protocol technically sound and planned in a manner that will lead to a meaningful outcome and allow testing the stated hypotheses?

Reviewer #1: Yes

Reviewer #2: Yes

3. Is the methodology feasible and described in sufficient detail to allow the work to be replicable?

Reviewer #1: No

Reviewer #2: Yes

4. Have the authors described where all data underlying the findings will be made available when the study is complete?

Reviewer #1: Yes

Reviewer #2: No

5. Is the manuscript presented in an intelligible fashion and written in standard English?

Reviewer #1: Yes

Reviewer #2: Yes

6. Review Comments to the Author

You may also provide optional suggestions and comments to authors that they might find helpful in planning their study.

Reviewer #1: In this manuscript Chacón-Díaz et al clearly showed the protocol of PEruvian Registry of ST-segment Elevation Myocardial Infarction II (PERSTEMI-II) study. Epidemiology of STEMI, outcome and follow-up data of STEMI have been already investigated in many European, North American and Asiatic manuscripts but few updated data are available for developing countries, in particular for Perù therefore this study can add updated scientific evidence in this field. In this manuscript the authors illustrated exhaustively the study objectives, the study design, statistical analysis with good abstract and complete references. No conflict of interest were reported neither discrepancies by ethical protocol.

Some observations should be made:

- In the “introduction” in hospital mortality data on STEMI for Europe and US are quite high because they refer to unselected patients, conversely for patients underwent to coronary angiography or coronary angioplasty in hospital mortality rate is lower. I suggest to add “unselected patients” in the text.

- In the “introduction” the data on STEMI reperfusion in Europe and North America are missing, I suggest to show it because they are very different from Peruvian data.

- In “outcome measure” more details should be add

- The strengths and limitation of the study are reported in the abstract but not in the discussions, they should be add in the main text

- In the “study desing and setting” the complete list of hospitals should be add in the a appendix

- In the “references” number 5 and number 9 should be updated with new ESC and ACC guidelines

I congratulate with the authors for trial design hoping in positive and significant results.

Reviewer #2: Dear authors,

Congratulations for this study protocol. It can provide valuable insights about the MI care in Peru. Although the protocol is interesting and potentially feasible, I have some comments to be addressed, below:

1) Abstract: in the Methods section, no description of the target population (age, demographics, sites, location, period). Please amend, accordingly, with information relevant to the understanding of the protocol.

2) Intro: some lines should ideally be added, about the quality criteria in MI care, as published periodically by the AHA/ACC (e.g.: Circulation: Cardiovascular Quality and Outcomes. 2017;10:e000032)..

3) Objective: if the authors aim to evaluate predictors of mortality among MI patients, this should be included among the aims of the study.

4) Stury procedures: a table with the variables that will be systematically collected would be helpful to the reader. Or maybe a flowchart of the study protocol, including such variables.

5) Outcome measures: please provide definitions for the measure "heart failure", as there are several ways to evaluate/define it.

6) Startistical analysis: please clearly define the study groups. From the text, I assume that the Cox models with be run for patients with and without reperfusio, but it is not clear at all in this section.

7) For the analysis of predictors of reperfusion, the plan is a multivariate regression analysis (logistic regression), right? Please clarify.

7. PLOS authors have the option to publish the peer review history of their article (what does this mean?). If published, this will include your full peer review and any attached files.

Reviewer #1: No

Reviewer #2: **Yes: **Bruno Ramos Nascimento, MD, MSc, PhD, FACC

---

## [Author Response · Author response to Decision Letter 0]

2 Sep 2021

September 1th, 2021

Dear Chiara Lazzeri Prof.

Academic Editor

PLOS ONE

Ref: Submission [PONE-D-21-11641]

Title: "Study protocol of the PEruvian Registry of ST-segment Elevation Myocardial Infarction II (PERSTEMI-II) study"

We thank the reviewers and the Academic Editor for their helpful comments and suggestions provided for our study protocol. All comments and suggestions have been addressed in the new revised version of the study protocol. All the comments and changes are described below according to the revision process. 

Journal Requirements

Response. Thank you for the recommendation. We have reviewed the manuscript, and it now follows the guidelines mentioned above.

Reviewer #1

In this manuscript Chacón-Díaz et al clearly showed the protocol of PEruvian Registry of ST-segment Elevation Myocardial Infarction II (PERSTEMI-II) study. Epidemiology of STEMI, outcome and follow-up data of STEMI have been already investigated in many European, North American and Asiatic manuscripts but few updated data are available for developing countries, in particular for Perù therefore this study can add updated scientific evidence in this field. In this manuscript the authors illustrated exhaustively the study objectives, the study design, statistical analysis with good abstract and complete references. No conflict of interest were reported neither discrepancies by ethical protocol. Some observations should be made:

1. In the “introduction” in hospital mortality data on STEMI for Europe and US are quite high because they refer to unselected patients, conversely for patients underwent to coronary angiography or coronary angioplasty in hospital mortality rate is lower. I suggest to add “unselected patients” in the text.

Response. Thank you for the recommendation. Since the reported incidence comes from studies about hospital admitted patients, we have added “in hospital admitted patients” to the text. 

2. In the “introduction” the data on STEMI reperfusion in Europe and North America are missing, I suggest to show it because they are very different from Peruvian data.

Response. Thank you for the recommendation. We have added the following information: “Regarding the use of reperfusion in STEMI, data from the CRUSADE registry mention that the percentage of reperfusion in the United States is 82.5%, and up to 7.2% do not receive reperfusion despite being indicated [10], while in Europe, it has recently been reported that reperfusion in STEMI is predominantly by percutaneous coronary surgery (72.2%) followed by fibrinolysis (18%), and 9% do not receive reperfusion [13].”

3. In “outcome measure” more details should be add

Response. Thank you for the recommendation. We added the following information. Now, it states: 

In-hospital mortality (cardiovascular and noncardiovascular).

Mortality at 30 days (cardiovascular and noncardiovascular).

One-year cardiovascular Mortality. 

In-hospital symptomatic heart failure (heart failure symptoms and/or signs [dyspnea, pulmonary rales, peripheral edema, third and fourth heart sounds] associated or not with a reduced ejection fraction less than 40%). 

4. The strengths and limitation of the study are reported in the abstract but not in the discussions, they should be add in the main text

Response. Thank you for the recommendation. We have added the following information: “Among the strengths of this study, the observational design will allow the inclusion of a large sample of patients, which will significantly contribute to the knowledge based on STEMI in Peru. This study is the first to examine the clinical-epidemiological characteristics of STEMI in high-resolution hospital centers in Peru with follow-up one year after the event, which allows us to learn about providing relevant information regarding these observable characteristics in daily clinical routine. Furthermore, the multicenter nature of the study will increase the external validity of the findings. However, regarding the limitations of the study, the observational design of the study can only describe associations and not causality. Moreover, there could be imprecision in the data analyzed since data from medical records will be used.”

5. In the “study design and setting” the complete list of hospitals should be add in the appendix

Response. Thank you for the recommendation. Since the Journal’s guidelines establish that manuscripts can be any length and the revised protocol has no long extension, we consider that this information could be included as part of the body of the manuscript.

6. In the “references” number 5 and number 9 should be updated with new ESC and ACC guidelines

Response. Thank you for the recommendation. The information and the references have been updated.

Reviewer #2

Dear authors, Congratulations for this study protocol. It can provide valuable insights about the MI care in Peru. Although the protocol is interesting and potentially feasible, I have some comments to be addressed, below:

1. Abstract: in the Methods section, no description of the target population (age, demographics, sites, location, period). Please amend, accordingly, with information relevant to the understanding of the protocol.

Response. Thank you for the recommendation. We have included the requested information. Now the Methods section in the Abstract includes the following information: “Methods and analysis: This will be a prospective, observational, multicenter study, with baseline and two follow-up assessments: at admission to the health service, 30 days and 12 months after admission. This multicenter study will be conducted in 27 hospitals located in the main cities of Peru. The patients included in the study will be over 18 years of age, of either sex and will have been admitted to the health facility with a diagnosis of acute coronary syndrome with ST-segment elevation.”

2. Intro: some lines should ideally be added, about the quality criteria in MI care, as published periodically by the AHA/ACC (e.g.: Circulation: Cardiovascular Quality and Outcomes. 2017;10:e000032).

Response. Thank you for the recommendation. We have added information about the quality of myocardial infarction care in the introduction section. It now reads: “It should be noted that the quality of hospital care received by patients with STEMI or non-ST-segment elevation myocardial infarction (NSTEMI) can reduce a large proportion of disparities in access to treatment, time to intervention, and cardiac rehabilitation. In this regard, updates have been made on assessments of hospital quality of care measures, highlighting seven measures (four of clinical effectiveness and three of patient safety) that address STEMI and NSTEMI patients. Particularly, in comatose STEMI patients with out-of-hospital cardiac arrest (as one of the specific clinical effectiveness measures) and in patients with cardiac arrest due to ventricular tachycardia and ventricular fibrillation, therapeutic hypothermia is considered and should be initiated as soon as possible. Regarding quality measures related to patient safety, it is observed that inappropriate use of nonsteroidal anti-inflammatory drugs, prasugrel at discharge in patients with a previous history of stroke or transient ischemic attack (TIA), and high doses of aspirin with ticagrelor at discharge, are measures that should be advised in the management of patients with both types of myocardial infarction and may reduce the risks of treatment failure.”

3. Objective: if the authors aim to evaluate predictors of mortality among MI patients, this should be included among the aims of the study.

Response. Thank you for the observation. We have added the following information to clarify the study aim “Likewise, the aim is to measure the incidence of STEMI complications and identify the main complications and to determine the incidence and risk factors of cardiovascular adverse events derived from STEMI at 30 days and 1 year of follow-up.” 

4. Study procedures: a table with the variables that will be systematically collected would be helpful to the reader. Or maybe a flowchart of the study protocol, including such variables.

Response. Thank you for the recommendation. We have prepared a table with a detailed description of the variables to be collected, which will be provided in the Supporting information Table (S1 Table).

5. Outcome measures: please provide definitions for the measure "heart failure", as there are several ways to evaluate/define it.

Response. Thank you for the recommendation. We added information about the measurement of heart failure. Now, it states “In-hospital symptomatic heart failure (heart failure symptoms and/or signs [dyspnea, pulmonary rales, peripheral edema, third and fourth heart sounds] associated or not with a reduced ejection fraction less than 40%).

6. Statistical analysis: please clearly define the study groups. From the text, I assume that the Cox models with being run for patients with and without reperfusion, but it is not clear at all in this section.

Response. Thank you for the recommendation. We have proceeded to clarify that the Cox models are run according to the type of reperfusion.

7. For the analysis of predictors of reperfusion, the plan is a multivariate regression analysis (logistic regression), right? Please clarify.

Response. Thank you for the recommendation. We have clarified the analysis of risk factors for successful reperfusion and adverse cardiovascular events. In addition, the crude and adjusted values were changed to bivariate and multivariate, respectively, to obtain relative risks because it is a prospective observational study. For further clarification, changes were made to the abstract and methods as follows:

In the Abstract: “Subsequently, to evaluate the risk factors for successful reperfusion and cardiovascular adverse events, generalized linear models of the binomial family with log link function will be used to estimate the bivariate and multivariate relative risk (RR) with their respective 95% confidence intervals.

In Methods: On the other hand, to evaluate the risk factors for successful reperfusion and cardiovascular adverse events, bivariate analysis of categorical variables will be performed using the chi-square test and for numerical variables the Student's t-test or Mann-Whitney U-test will be used according to compliance with the normal distribution assumption. Likewise, generalized linear models of the binomial family with log link function will be used to estimate the bivariate and multivariate relative risk (RR) and their respective 95% CI. 

Sincerely,

The Authors.

---

## [Editor Report · Decision Letter 1]

7 Sep 2021

Study protocol of the PEruvian Registry of ST-segment Elevation Myocardial Infarction II (PERSTEMI-II) study

PONE-D-21-11641R1

Dear Dr. Vargas-Fernández,

We’re pleased to inform you that your manuscript has been judged scientifically suitable for publication and will be formally accepted for publication once it meets all outstanding technical requirements.

Kind regards,

Chiara Lazzeri

Academic Editor

PLOS ONE
---

## [Editor Report · Acceptance letter]

10 Sep 2021

PONE-D-21-11641R1 

Study protocol of the PEruvian Registry of ST-segment Elevation Myocardial Infarction II (PERSTEMI-II) study 

Dear Dr. Vargas-Fernández:

I'm pleased to inform you that your manuscript has been deemed suitable for publication in PLOS ONE. Congratulations! Your manuscript is now with our production department. 

Kind regards, 

on behalf of

Dr. Chiara Lazzeri 

Academic Editor

PLOS ONE